# A Cost Assessment of Tree Plantation Failure under Extreme Drought Events in France: What Role for Insurance?

Sandrine Brèteau-Amores [1,*], Marielle Brunette [1] and Pablo Andrés-Domenech [2]

1 Université de Lorraine, Université de Strasbourg, AgroParisTech, CNRS, INRAE, BETA, 54000 Nancy, France
2 AgroParisTech, BETA, 54000 Nancy, France
* Correspondence: sandrine.breteauamores@gmail.com

**Abstract:** Research Highlights: We analyze the costs of plantation failure and evaluate the distribution of replantation costs and risk sharing between the forestry company and the forest owner in France. Background and Objectives: Due to the lack of a clear definition of drought, forestry companies are increasingly considered as liable for plantation failure, increasing their costs and leading to financial instability. In this context, this paper aims to address the following questions. In the case of plantation failure, is it less costly to replant, not replant, or restart the whole plantation? What is the impact of changing the liability scheme between the company and the forest owner in terms of replantation costs and risk sharing? Materials and Methods: We performed a cost assessment of different itineraries of plantations as a function of different mortality rates. The breakdown of the replantation costs between the company and the forest owner was also investigated. Results: No replanting is the least expensive option for the forest owner, followed by replanting and then by starting the whole plantation anew. Reducing the company's liability is an interesting option to reduce its exposure to risk. Conclusions: Modifications of the company's liability allows for the inclusion of private insurance contracts against plantation failure.

**Keywords:** forest; regeneration; plantation; drought; insurance; costs





## 1. Introduction

Forest regeneration is the basis of forest renewal, which is essential for the sustainable management of the forest resource. This regeneration can be performed by natural regeneration or plantation. Even in the case of natural regeneration, planting may be necessary to enrich (i.e., increase diversity or density) or improve (i.e., to better adapt it to the environment) the stand. Indeed, diversifying the stand or substituting the current species with another one (better adapted or more productive) requires planting. However, the seedling stage is known to be the most vulnerable phase in the life cycle of a forest stand. In addition to normal planting pressure (transplantation shock because the plant cannot adapt to its new environment), these young plants may be subject to various stresses, including: abiotic (e.g., frost, hail, high temperatures, drought); biotic (attacks by insects and fungi, which are often specific to very young trees that may carry these risks with them and contribute to spread them when they leave the nursery, tree competition, grazing); or anthropogenic (inappropriate soil work, preparation or storage of seedlings, planting or other silvicultural operations).

In the context of changing climate conditions, seedlings are exposed to additional or increased stresses. Many ecological models forecast repercussions that include more significant disturbances, including forest fires, insect infestations, changes in the distribution of various tree species, changes in species' growth rates, and droughts [1]. Such alterations would have significant ecological and economic repercussions, ranging from altered ecosystem-service flows to diminished economic value [2].

In France, the dry summers of 2018, 2019, and 2020 (with each subsequent year worse than the previous one) caused significant damage to the forests and decreased planting

success [3]. Almost 89% of the plants died due to abiotic causes of which 60% perished due to drought [3]. Indeed, drought periods are becoming more intense and frequent [4], resulting in a large growth/productivity reduction and weakening both forest health and the forest-wood industry.

Forest plantations are consequently critical for forest renewal towards the end of these crises. The success of this phase is determined by the survival of the young plants. In France, plantations are carried out by a variety of forestry companies (in charge of nursery and forest management operations), which generally provide insurance (i.e., a warranty service). This insurance is based on the observation of the success of the plantation after the first growing season. In France, in the case of plantation failure, the forestry company provides a replacement for dead plants if the mortality exceeds 20% of the plantation. Mortality not covered by the insurance (20% in this case) remains the forest owner's liability. However, the insurance can be waived if the mortality of the plants is unrelated to the quality of the forestry company's service, for example, in periods of excessive drought or damage caused by animals, including intense browsing or infestations. Nevertheless, due to the lack of a clear and uniform characterization of drought events, this insurance clause is rarely applied, and plantation failure is more often than not the forestry company's liability, greatly increasing its costs. In view of the reforestation sector's difficulties in reducing the costs related to excess mortality, changing the clause governing insurance services for the success/survival of plantations in the event of extreme drought episodes can allow the forestry company to be released from its liability.

In this context, two main questions emerge: (i) in the case of a drought shock in the plantation, is it better, in terms of costs for the forest owner, to replant, not to replant, or to fully replant (i.e., starting the whole plantation from scratch)? (ii) What is the impact of changing the liability scheme between the forestry company and the forest owner in terms of risk sharing and replantation costs? In this paper, we propose answers to these two questions.

The existing literature on plantation failure due to drought is very limited. On the one hand, the question of plantation failure has been mainly addressed to measure the impact of game on silvicultural practices (see the study of Barrere et al. [5] as a recent example). On the other hand, many studies have investigated species' drought tolerance/resistance on saplings and seedlings due to the ease of experimentation at young stages (see the literature review by Grossnickle [6] and a recent study by Pardos and Calama [7] as examples). Regarding the literature in economics, to the best of our knowledge, only Morkovina et al. [8] have analyzed the economic aspects of forest regeneration. They compared two types of planting (traditional and innovative) through the discounted return on investment, i.e., the net present value of gains divided by the net present value of costs. They show that the innovative technology (seedlings grown with closed root systems) requires minimum costs for growing. Moreover, an increasing number of rotations with the innovative technology leads to a decrease in the production costs and makes seedlings of English oak competitive in price with seedlings grown with the traditional technology.

This short literature review shows that the literature is very narrow on the topic of plantation failure due to drought. This observation allows us to highlight two main gaps in the literature that we try to fill. First, there is no study to date that assesses the economic aspects of plantation failure due to drought, whereas precise information on the costs associated with such failure seems necessary to improve the decision-making process of forestry companies and forest owners. Second, the issue of shared liability between the forest owner and the forestry company in case of plantation failure due to drought has not yet been investigated, even though the viability of the forestry companies is at stake.

In this article, we analyzed the costs of plantation failure and evaluated the distribution of these costs between the forestry company and the forest owner in France. To do this, we proceeded in three steps. First, we performed a cost assessment of different itineraries of plantations for the three main planted species in France: maritime pine (*Pinus pinaster* Ait.), sessile oak (*Quercus petraea* Liebl.), and Douglas fir (*Pseudotsuga menziessi* Mirb.). This

allowed us to identify the itinerary that minimized the plantation costs for each species. Second, based on this chosen itinerary for each species, we evaluated and identified the best option in the event of plantation failure between: (i) replanting/replacing the dead plants with new ones; (ii) no replanting; and (iii) full replanting. These three options were compared for different mortality rates. Third, we investigated the breakdown of the replantation costs and thus the share of risk between the forestry company and the forest owner by considering different deductible levels and under different mortality rates.

In this way, we contribute to the literature in economics on this topic by: (i) providing a cost assessment of plantation failure and (ii) considering risk sharing between the forestry company and the forest owner. Our results show the importance of including the private insurance sector in the coverage of plantation failure.

The rest of the paper is structured as follows. The materials and the methods are presented in Section 2. Section 3 provides the results. The results are then discussed in Section 4, and Section 5 concludes.

## 2. Materials and Methods

### 2.1. Materials

2.1.1. Description of the French Context (https://franceboisforet.fr/wp-content/uploads/2021/04/Brochure_chiffresClesForetPrivee_2021_PageApage_BD.pdf (accessed on 3 February 2023) and [9])

Forests represent one third of the total surface in France. Three quarters of these forests are private, which represent 12.7 Mha owned by 3.3 million owners. The remaining quarter (4.3 Mha) is public and is divided between state-owned forests (1.5 Mha) and other public forests (2.8 Mha), mainly communal forests.

The stock of standing timber has continuously increased since 1985 (1.8 to 2.8 billion $m^3$ over the period 1985–2022), mainly due to the increase in forest area (abandonment of agricultural land). However, this upward trend has been partially hindered by the lower biological production and increasing removals and mortality (11.4 $Mm^3$/year on average over the period 2012–2020).

2.1.2. Species of Interest

We restricted our computations to the three main species used for plantations in France according to the statistics (https://agriculture.gouv.fr/statistiques-annuelles-sur-les-ventes-de-graines-et-plants-forestiers (accessed on 3 February 2023)) on sales of forestry seeds and seedlings during the period 2018–2021: maritime pine (*Pinus pinaster* Ait.), Douglas fir (*Pseudotsuga menziessi* Mirb.), and sessile oak (*Quercus petraea* Liebl.).

Maritime pine is native to northern Africa and southern Europe and was heavily planted in southwest France in the mid-19th century. Maritime pine is the most commonly planted species and accounts for 140 $Mm^3$ and 1.06 Mha in France [9]; some 15,000 ha of improved seedlings of maritime pine are annually planted in southwest France. Well-adapted to dry to semi-arid conditions, maritime pine is characterized by high productivity and drought resistance and tolerance in Mediterranean regions [10,11]. Along its natural distribution range, maritime pine grows under a wide range of climatic conditions with and without water stress, thus providing a high intra-population variability [12]. This variability allows it to resist a wide range of responses to environmental pressures such as extreme drought events [13].

Native to North America, Douglas fir was heavily introduced in France during the second half of the 20th century. France is the main Douglas fir producer in Europe outside of its region of origin [14], with 135 $Mm^3$ of growing stock and 426,000 ha [9]. Douglas fir is valued by foresters for its rapid growth and the quality of its wood [15]. However, it is more sensitive to high temperatures due to its high leaf area than to droughts. In addition to this, whereas Douglas fir is sometimes considered as a drought-resistant species [16], it does not seem to be well-adapted to the accumulation of intense and recurrent episodes of drought after a severe one, which is often characterized as a lack of resilience.

The most common and significant deciduous wood species in France is sessile oak. Along with pedunculate oak (*Quercus robur* L.), it accounts for 231 Mm$^3$ of growing stock, 1.76 Mha [9], and a high commercial interest. Sessile oak is more resistant to soil water shortage and competition than pedunculate oak [17]. Silvicultural interventions are often necessary to improve the survival and growth rate of oaks [18]. For instance, protection against herbivorous game might be needed to ensure their successful regeneration [17].

2.1.3. Itineraries of Plantations

As mentioned above, we considered three species: maritime pine (MP), sessile oak (SO), and Douglas fir (DF). For each species, we considered various possible types of planting: container seedlings (CS) planted with a cane (C) or in individual pits (IP), and seedlings supplied with bare roots (SBR) and planted with a pickaxe (P) or in individual pits. We also considered different types of game protection: fencing (F), individual protection (IGP), and repellent (R). This resulted in 14 different itineraries for plantation; two for maritime pine, four for Douglas fir, and eight for sessile oak, as follows (Table 1):

**Table 1.** Itineraries of plantations with their corresponding code and their detailed meaning. The itinerary code (IT code) corresponds to the species (maritime pine (*Pinus pinaster* Ait.), Douglas fir (*Pseudotsuga menziessi* Mirb.), and sessile oak (*Quercus petraea* Liebl.)), followed by the type of planting and the type of game protection.

| IT Code | Explanation |
| --- | --- |
| MP_CS-C | Maritime pine: container seedlings planted with a cane. |
| MP_CS-C+R | Maritime pine: container seedlings planted with a cane + game repellent. |
| DF_CS-C+R | Douglas fir: container seedlings planted with a cane + game repellent. |
| DF_CS-IP+R | Douglas fir: container seedlings planted in individual pits + game repellent. |
| DF_SBR-IP+R | Douglas fir: seedlings supplied with bare roots and planted in individual pits + game repellent. |
| DF_SBR-P+R | Douglas fir: seedlings supplied with bare roots and planted with a pickaxe + game repellent. |
| SO_CS-C+F | Sessile oak: container seedlings planted with a cane + fencing. |
| SO_CS-C+ IGP | Sessile oak: container seedlings planted with a cane + individual protection. |
| SO_CS-IP+F | Sessile oak: container seedlings planted in individual pits + fencing. |
| SO_CS-IP+IGP | Sessile oak: container seedlings planted in individual pits + individual protection. |
| SO_SBR-P+F | Sessile oak: seedlings supplied with bare roots and planted with a pickaxe + fencing. |
| SO_ SBR-P+IGP | Sessile oak: seedlings supplied with bare roots and planted with a pickaxe + individual protection. |
| SO_SBR-IP+F | Sessile oak: seedlings supplied with bare roots and planted in individual pits + fencing. |
| SO_SBR-IP+IGP | Sessile oak: seedlings supplied with bare roots and planted in individual pits + individual protection. |

The data for the plantation itineraries with their associated costs were provided by the forest cooperatives, "Forêt & Bois de l'Est" and "Alliance Forêts Bois". An example is presented in Table 1 for maritime pine (see the other itineraries in Table A1 Appendix A).

The operations that take place in the different itineraries are those indicated in Table 2.

- Preparation of the vegetation and the soil. This is the first step in a silvicultural operation chain during a commercial forest rotation. It is carried out as a part of the actions for forest regeneration after the final cut. The goal of soil preparation is to enhance seed germination and create favorable conditions for seedling growth.
- Plant + plantation. In this step, it is possible to choose between plant seeds or seedlings. The advantage of seedlings is that they already have some growth and compete better with surrounding vegetation. Additionally, the selection of genetically adapted material is strongly recommended in order to improve the survival of the plantation.
- Game protection. This includes fencing, individual protection, or repellent. In this step, some protection is offered to young plants to prevent them from being eaten by herbivores such as game or insects. Fencing consists of enclosing the entire perimeter of the plantation with fences resistant to both domestic and wild livestock. Individual fencing consists of protecting each of the young plants. The repellent method is used to protect the seedlings against direct attacks and is applied to each plant.

- Replantation. This step may take place in the second year if/when the dead seedlings are replaced by new plants.
- Removal (or maintenance). Unwanted vegetation growing in the planting area (or in the spacing lines) is removed to improve seedling growth. This step could take place in the second and third year after the first plantation.

**Table 2.** Itinerary of plantations for maritime pine with the detailed operations associated with their year of realization and their costs (in EUR/ha). The code of the itinerary (IT code) corresponds to: (i) the species (MP for maritime pine (*Pinus pinaster* Ait.)), (ii) the type of planting (CS-C for container seedlings planted with a cane), and (iii) the type of protection against game (none or R for game repellent).

| Species | IT Code | Year | Operation | Stand Density | Price per Unit | Cost per ha |
|---|---|---|---|---|---|---|
| Maritime pine | MP_CS-C | 1 | Preparation of the vegetation and the soil | | | €600 |
| | | 1 | Plant + plantation | 1350 | €0.53 | €716 |
| | | 2 | Replantation: Plant + plantation | | €0.78 | |
| | | 3 | Maintenance | | | €300 |
| | | 7 | Maintenance | | | €100 |
| | MP_CS-C+R | 1 | Preparation of the vegetation and the soil | | | €600 |
| | | 1 | Plant with game repellent + plantation | 1350 | €0.63 | €851 |
| | | 2 | Replantation: Plant + plantation | | €0.78 | |
| | | 3 | Maintenance | | | €300 |
| | | 7 | Maintenance | | | €100 |

### 2.2. Cost Analysis

We performed a cost analysis of plantation failure in three steps. First, we assessed the costs for each itinerary in order to identify the one that minimized the plantation costs for each species. Second, based on this chosen itinerary, we compared the three options for each species: (i) replanting (i.e., replacing the dead plants by new ones); (ii) no replanting; and (iii) full replanting (i.e., restarting a new plantation from scratch). Finally, we investigated the share of the risk between our two stakeholders: the forest owner and the forest company.

#### 2.2.1. Assessment of the Costs for Each Itinerary

In order to estimate the economic losses of plantation failure, we used the average price of the lowest diameter class from the "Comptes de la Forêt" of the Observatory for Forest Economics (https://beta-economics.fr/platforms/olef-platform/ (accessed on 3 February 2023)) (OLEF, BETA, France). The annual tree growth at the young stages was estimated based on the growth yield tables in the silvicultural guides proposed by the French National Forest Office (ONF). The values of both parameters for each species are presented in Table 3.

**Table 3.** Annual tree growth and mean price for the lowest diameter class for the three species (maritime pine (*Pinus pinaster* Ait.), Douglas fir (*Pseudotsuga menziessi* Mirb.), and sessile oak (*Quercus petraea* Liebl.)).

| Species | Annual Tree Growth (m³/Year) | Mean Price (€/m³) |
|---|---|---|
| Maritime pine | 0.0012 | 15 |
| Douglas fir | 0.0085 | 28 |
| Sessile oak | 0.0025 | 15 |

We then computed the loss of dead plants as follows:

$$Loss = tree\ volume \times stand\ density \times stand\ mortality\ rate \times mean\ wood\ price, \quad (1)$$

where the total monetary loss is computed as the product of the total standing volume, the mortality rate (i.e., the stand density loss), and the mean wood price (see Table 3).

The tree volume can then be estimated as:

$$Tree\ volume = age \times annual\ tree\ growth. \quad (2)$$

In order to perform a comparison of the itineraries based on the minimization of costs, we computed the total discounted cost (TDC) of the different itineraries as follows:

$$TDC = \sum_{n=0}^{N} \frac{C_n}{(1+r)^n}, \quad (3)$$

where $C$ are the costs, $r$ is the discount rate, $n$ is the stand's age, and $N$ is the plantation establishment period. For our computations, we used a discount rate of 2%, rate commonly applied in forestry (after a discussion with forestry experts). The total costs C are the sum of the plantation costs (from Tables 2 and A1), and the loss by mortality (Equation (1)) and the replantation costs (from Tables 2 and A1) when applicable. We assumed here that the forest owner's objective is to minimize the total discounted costs of the plantation.

We first computed total discounted costs for each itinerary without mortality (and therefore without any replantation) in order to identify the least expensive itinerary for each species.

### 2.2.2. Assessment of Replanting, not Replanting, and Restarting the Whole Plantation

For the sake of simplicity, we considered the least expensive itinerary to assess and compare the three scenarios by species: no replantation of the dead plants, replantation of the dead plants, and full replanting (i.e., starting the whole plantation anew).

Indeed, we then computed the total discounted costs of these chosen itineraries considering a mortality rate ranging from 0 to 90%. The range of mortality rate that we assumed represents the increase in risk associated with extreme drought events.

### 2.2.3. Risk Sharing Analysis between the Forest Owner and the Forest Company

We emphasized the analysis of the case of replantation by looking at the breakdown of the replantation costs between the forestry company and the forest owner.

Total discounted costs of the replantation scenario for each species were computed considering different deductible levels: 20% (current practice), 50%, and 80%. By doing this, we considered the share of risk borne by each of the two stakeholders. Indeed, the deductible is the part of risk (dead plants) that is the forest owner's liability, whereas the other part is the forestry company's liability.

More precisely, the current practice in France is to have a deductible of 20%. This means that in the event of plantation failure, the forestry company provides a replacement for dead plants whenever mortality exceeds 20% of the plantation and only for any excess mortality beyond 20%. Mortality not covered by the insurance (20% in this case) remains the forest owner's liability. Here, the deductible of 20% implies that the forestry company bears most of the risk.

Increasing the deductible means a lower liability for the forestry company and more liability for the forest owner. This is why we tested two other deductible levels that imply a higher risk borne by the forest owner: 50% and 80%.

Finally, we compared the replantation costs pertaining to the forestry company and the total costs pertaining to the forest owner for these three deductible levels in order to investigate the impact of changing this level for each stakeholder.

## 3. Results

### 3.1. Comparison of the Total Discounted Costs of Different Plantation Itineraries for the Three Species

Table 4 presents the total discounted costs borne by the forest owner for all the itineraries considered and for each species. The results show that considering game protection (regardless of the type (fencing (F), individual protection (IGP) or repellent (R)) or plantation of bare root plants (SBR) increases the planting costs for all the species.

**Table 4.** Total discounted costs borne by the forest owner ranked from the least to the most expensive itinerary for each species. The first one appears in bold in EUR/ha and the others are expressed as a variation in percentage according to the least expensive itinerary. The itinerary code corresponds to the species (MP for maritime pine (*Pinus pinaster* Ait.), DF for Douglas fir (*Pseudotsuga menziessi* Mirb.), SO for sessile oak (*Quercus petraea* Liebl.)), followed by the type of plants (CS for container seedlings, SBR for seedlings supplied with bare roots), the plantation type (C for plantation with a cane, P with a pickaxe, IP in individual pits), and game protection (F for fencing, IGP for individual protection, R for game repellent).

| Species | IT Code | Total Discounted Costs (€/ha) |
|---|---|---|
| Maritime pine | MP_CS-C | €1659/ha |
| | MP_CS-C+R | +8% |
| Douglas fir | DF_CS-IP+R | €4213/ha |
| | DF_CS-C+R | +2% |
| | DF_SBR-IP+R | +6% |
| | DF_SBR-P+R | +10% |
| Sessile oak | SO_CS-IP+F | €8546/ha |
| | SO_CS-C+F | +2% |
| | SO_SBR-IP+F | +6% |
| | SO_SBR-P+F | +10% |
| | SO_CS-IP+IGP | +12% |
| | SO_SBR-IP+IGP | +18% |
| | SO_CS-C+IGP | +26% |
| | SO_SBR-P+IGP | +34% |

More precisely, adding game protection increases the costs by 8% for maritime pine, so that MP_CS-C < MP_CS-C+R for the forest owner.

For Douglas fir, the cost ranking (in absolute terms) is as follows: DF_CS-IP+R < DF_CS-C+R < DF_SBR-IP+R < DF_SBR-P+R. It is worth noting that the plants for the container seedlings (CS) cost less than those supplied with bare roots (SBR), and planting in individual pits (IP) decreases the costs. Although the preparation of the vegetation and the soil is significantly more expensive for plants in individual pits than those planted with a cane (C), the cost is compensated by the lower planting density (1300 vs. 1600). That is why DF_CS-IP+R is the least expensive itinerary of the four.

The same explanation applies to the sessile oak itineraries analyzed: container seedlings (CS) and planting in individual pits (IP) are less costly (when all costs and not just plantation costs are accounted for) than the other alternatives. In addition to this, fencing (F) is less expensive than individual protection (IGP). Consequently, SO_CS-IP+F is the least expensive itinerary.

### 3.2. Comparison of the Total Discounted Costs between Replanting, not Replanting, and Restarting the Whole Plantation

Since our objective is to minimize the cost, we only considered the least expensive itinerary in this section for each species in Table 4, i.e., the three in bold (MP_CS-C, DF_CS-IP+R, SO_CS-IP+F).

First, we present the total discounted costs that apply to the forest owner for each species for different mortality rates (from 0 to 90%) considering three scenarios: no re-

plantation after a drought shock, replantation with and without insurance by the forestry company, and full replanting. When considering insurance, we assume three distinct deductible levels: 20% (the status quo), 50%, and 80%. Table 5 presents the corresponding results for maritime pine (see Tables A2 and A3 for Douglas fir and sessile oak, respectively, in Appendix B).

**Table 5.** Total discounted costs borne by the forest owner in EUR/ha for maritime pine (*Pinus pinaster* Ait.) (itinerary "MP_CS-C") considering no replantation after drought shock ("Without RP"), replantation without insurance by the forestry company ("With RP and no INS"), replantation with insurance by the forestry company ("With RP and INS") with three deductibles ("D 20%", "D 50%", "D 80%"), and full replanting ("New plantation"), for different mortality rates.

| Mortality Rate | Without RP | With RP and no INS | With RP and INS (D 20%) | With RP and INS (D 50%) | With RP and INS (D 80%) | New Plantation |
|---|---|---|---|---|---|---|
| 0 | €1659 | €1659 | €1659 | €1659 | €1659 | |
| 0.1 | €1664 | €1765 | €1765 | €1765 | €1765 | €2954 |
| 0.2 | €1669 | €1871 | €1871 | €1871 | €1871 | €2959 |
| 0.3 | €1673 | €1977 | €1876 | €1977 | €1977 | €2963 |
| 0.4 | €1678 | €2083 | €1881 | €2083 | €2083 | €2968 |
| 0.5 | €1683 | €2189 | €1885 | €2189 | €2189 | €2973 |
| 0.6 | €1687 | €2295 | €1890 | €2194 | €2295 | €2977 |
| 0.7 | €1692 | €2401 | €1895 | €2198 | €2401 | €2982 |
| 0.8 | €1697 | €2507 | €1899 | €2203 | €2507 | €2987 |
| 0.9 | €1702 | €2612 | €1904 | €2208 | €2511 | €2991 |

Table 5 shows that full replanting ("New plantation") is always the costliest scenario for the forest owner, and no replantation ("Without RP") is always the best one, regardless of the mortality rate. The four scenarios with replantation are in between. Up to a mortality rate of 20%, the four scenarios are associated with the same costs. When the mortality rate increases beyond 20%, the scenario with insurance linked to the lowest deductible (20% in this case) becomes the least expensive for the forest owner. This result is directly associated with a lower liability level for the forest owner. In sum, the costs increase with the deductible and the replantation without insurance is the costliest of the four scenarios with replantation. Regarding the cost increase following increased mortality, note that full replanting has the lowest increase (1%) because its costs are not affected by replantation costs. No replantation is then the lowest one (2%), followed by replantation with (13, 25, and 34% for a deductible of 20, 50, and 80%, respectively) and without (36%) insurance. The same results are observed for Douglas fir and sessile oak.

Second, we emphasize the analysis by investigating the breakdown of the replantation costs (i.e., the share of risk) between the forestry company and the forest owner. Table 6 presents these results considering three different deductible levels and different mortality rates for maritime pine.

The results show that increasing the deductible level translates into a cost reduction for the forestry company and a cost increase for the forest owner. More precisely, moving from a deductible of 20% to 50% increases the forest owner's costs by 5–16% and decreases the forestry company's costs by 43–100%. Moving from 20% to a deductible of 80% increases the forest owner's costs by 5–32% and decreases the forestry company's costs by 86–100%. In relative terms, the cost reduction for the forestry company is greater than the cost increase for the forest owner, where a deductible of 50% has the smallest impact for the forest owner.

Tables 7 and 8 are equivalent to Table 6 for the other two species, Douglas fir and sessile oak, respectively. The same trends are observed for the two species with the same range of cost decrease for the forestry company. Concerning the cost increase for the forest owner, the range is lower for Douglas fir (4–12% and 4–24% for a deductible of 50% and 80%, respectively) and even more for sessile oak (3–10% and 3–19% for a deductible of 50%

and 80%, respectively) compared with maritime pine. A deductible of 50% has a lower impact for the forest owner in sessile oak plantations.

**Table 6.** Total discounted costs in EUR/ha for maritime pine (*Pinus pinaster* Ait.) (itinerary "MP_CS-C") for the forestry company and the forest owner for a deductible of 20% (current practice), 50%, and 80%. The costs are expressed (for different mortality rates) as variations in percentage of the total discounted costs with respect to the current practice (i.e., status quo).

| Deductible | 20% (status quo) | | 50% | | 80% | |
|---|---|---|---|---|---|---|
| Mortality Rate | Company | Forest Owner | Company | Forest Owner | Company | Forest Owner |
| 0 | €0 | €1659 | 0% | 0% | 0% | 0% |
| 0.1 | €0 | €1765 | 0% | 0% | 0% | 0% |
| 0.2 | €0 | €1871 | 0% | 0% | 0% | 0% |
| 0.3 | €101 | €1876 | −100% | +5% | −100% | +5% |
| 0.4 | €202 | €1881 | −100% | +11% | −100% | +11% |
| 0.5 | €304 | €1885 | −100% | +16% | −100% | +16% |
| 0.6 | €405 | €1890 | −75% | +16% | −100% | +21% |
| 0.7 | €506 | €1895 | −60% | +16% | −100% | +27% |
| 0.8 | €607 | €1899 | −50% | +16% | −100% | +32% |
| 0.9 | €708 | €1904 | −43% | +16% | −86% | +32% |

**Table 7.** Total discounted costs in EUR/ha for Douglas fir (*Pseudotsuga menziessi* Mirb.) (itinerary "DF_CS-IP+R") for the forestry company and the forest owner for a deductible of 20% (current practice), 50%, and 80%. The costs are expressed (for different mortality rates) as variations in percentage of the total discounted costs with respect to the current practice (i.e., status quo).

| Deductible | 20% (status quo) | | 50% | | 80% | |
|---|---|---|---|---|---|---|
| Mortality Rate | Company | Forest Owner | Company | Forest Owner | Company | Forest Owner |
| 0 | €0 | €4213 | 0% | 0% | 0% | 0% |
| 0.1 | €0 | €4423 | 0% | 0% | 0% | 0% |
| 0.2 | €0 | €4632 | 0% | 0% | 0% | 0% |
| 0.3 | €187 | €4654 | −100% | +4% | −100% | +4% |
| 0.4 | €375 | €4676 | −100% | +8% | −100% | +8% |
| 0.5 | €562 | €4698 | −100% | +12% | −100% | +12% |
| 0.6 | €750 | €4720 | −75% | +12% | −100% | +16% |
| 0.7 | €937 | €4742 | −60% | +12% | −100% | +20% |
| 0.8 | €1125 | €4764 | −50% | +12% | −100% | +24% |
| 0.9 | €1312 | €4786 | −43% | +12% | −86% | +23% |

**Table 8.** Total discounted costs in EUR/ha for sessile oak (*Quercus petraea* Liebl.) (itinerary "SO_CS-IP+F") for the forestry company and the forest owner for a deductible of 20% (current practice), 50%, and 80%. The costs are expressed (for different mortality rates) as variations in percentage of the total discounted costs with respect to the current practice (i.e., status quo).

| Deductible | 20% (status quo) | | 50% | | 80% | |
|---|---|---|---|---|---|---|
| Mortality Rate | Company | Forest Owner | Company | Forest Owner | Company | Forest Owner |
| 0 | €0 | €8546 | 0% | 0% | 0% | 0% |
| 0.1 | €0 | €8849 | 0% | 0% | 0% | 0% |
| 0.2 | €0 | €9152 | 0% | 0% | 0% | 0% |
| 0.3 | €294 | €9161 | −100% | +3% | −100% | +3% |
| 0.4 | €587 | €9170 | −100% | +6% | −100% | +6% |
| 0.5 | €881 | €9180 | −100% | +10% | −100% | +10% |
| 0.6 | €1175 | €9189 | −75% | +10% | −100% | +13% |
| 0.7 | €1468 | €9198 | −60% | +10% | −100% | +16% |
| 0.8 | €1762 | €9208 | −50% | +10% | −100% | +19% |
| 0.9 | €2055 | €9217 | −43% | +10% | −86% | +19% |

## 4. Discussion

The main results show that no replanting is less expensive for the forest owner than replanting or full replanting. In the case of replantation (i.e., replacing the dead trees), the forest owner's costs increase with the forestry company's deductible. More precisely, regarding the share of risk between the forest owner and the forestry company, increasing the deductible results in a higher cost reduction for the forestry company than the cost increase for the forest owner.

In this context, we identified two ways of solving the forestry company's problem of financial instability related to increasing costs due to plantation failure.

First, since the problem emerges due to a lack of clear and uniform characterization of drought events, having a reliable drought index may solve the problem. More precisely, one such index would make it possible to represent the intensity of droughts numerically. These indices can include different levels of complexity and qualify droughts in terms of meteorological (Standardized Precipitation Index based on precipitation), hydrological (Soil Water Index based on soil properties), or agricultural (Standardized Precipitation-Evapotranspiration Index based on a water balance that considers precipitation and temperature) conditions. As soon as the plantation failure is clearly identified as a consequence of a drought event (through a reliable index), the forestry company's liability will not be engaged. In that case, the costs associated with the plantation failure would be borne by the forest owner.

One could even imagine that insurance companies from the private sector propose contracts specifically aimed at plantation failure due to drought. In terms of costs, including clauses specific to this index in the replanting contracts that explicitly exclude the forestry company's liability would result in a reduction in the forestry company's costs. They would no longer be liable for drought-related issues. Only planting problems would fall within their liability. Consequently, this would lead to an increase in the forest owners' costs. Forest owners could also transfer the risk to the insurer by paying an insurance premium. In that case, a third actor enters the relationship.

Introducing a private insurer in the relationship between the forest owner and forestry company would be interesting to analyze, especially in terms of liability and risk sharing. Traditionally, the forest insurance contract insures fire and/or storm damages depending on the country [19–21]. A recent article considers drought insurance in forests [22]. To the best of our knowledge, private insurance for plantation failure in forests does not exist. However, further investigations are needed in the field of forest ecology in order to have this reliable index first; this implies having a good correlation between drought damage and the defined index.

Second, if it is not possible to have a reliable drought index and it then becomes impossible to disentangle plantation failure due to drought from other planting problems, our results reveal that the deductible level may be an interesting vector to use. Indeed, we show that increasing the deductible level makes it possible to substantially reduce the costs supported by the forestry company with a less-than-proportional increase of the forest owner's costs, where a deductible of 50% has the smallest impact for the forest owner. This scheme may also reveal a role for private insurance companies. In the case of a deductible of 50%, we can easily imagine that the forest owner would remain liable when the mortality rate is below 20%, the private insurer would take charge of the next 20–50% of plantation failure and, finally, the forestry company would insure the higher damage, above 50%, much like we would see in the agricultural sector (i.e., the reform of the French crop insurance scheme will take place from 1 January 2023. One of the proposals of the reform is a three-level system: 0–20% of the crop damage to be borne by the farmer, 20–50% by the crop insurance, and over 50% by the State [23]). It should be noted here that this could create a potential moral hazard problem linked to the increase in the deductible. Indeed, it could decrease the incentive for the forestry company to perform good-quality work and encourage it to adopt less conscientious practices. This point should be investigated in future research.

The two options that we considered here are not necessarily substitutable; rather they can be considered to be complementary in order to increase the financial stability of the forestry company. Indeed, considering both a reliable drought index and an increase in the deductible level could lead to a significant reduction in the forestry company's cost (while resulting in increasing costs for the forest owner).

Moreover, these two options highlight a potentially important role for the private insurance market. Our results show that no replanting was less expensive than replanting, which is the typical decision of forest owners whenever they experience low mortality (especially up to 20% of dead plants). Therefore, having the option to insure the plantation failure through a private insurance contract may increase the attractiveness of replantation as compared to no replantation.

This attractiveness is important when considering the maintenance of forest cover, as well as to avoid ecosystem instability (e.g., gaps due to dead plants can disturb light proportion in the forest canopy or temporarily leave the soil bare, thus causing it to dry out). Taking the services provided by forests into account as a way to tackle climate change challenges (carbon storage, substitution of fossil-fuel-based products by wood products) also points at the need to replant. Maintaining forest and facing risks increase cost: the increase in mortality due to extreme drought events induced an increase in plantation costs in our study. In addition to facing this climate crisis, forest owners have to deal with other ones. The current energy (fuel) and material (steel) crises hinder the production and the use of technical equipment for silvicultural operations and also induce a cost increase, making it difficult to implement forest management. Therefore, forest owners will need more support to be able to maintain forest health and restore their forests after a crisis.

Since we have discovered a potential role for the insurance market in the coverage of plantation failure, the question of the acceptability of such a contract by forest owners needs to be addressed. More precisely, it will be interesting to assess their willingness to pay for such an insurance. This certainly opens an avenue for future research.

Other perspectives can be considered to improve our study. First, we assume here that the itineraries are performed in a similar way, regardless of mortality. Although this will not change, qualitatively, our results (i.e., ranking of the scenarios), it should be noted that some forest operations (e.g., removal) might not be carried out in the case of high mortality (e.g., 80%, 90%) and no replanting.

Second, we performed a cost-based assessment of planting itineraries. However, this method could be improved by using a cost-benefit analysis. Thus, the analysis would consider both the total costs and the economic benefits that would result from the sale of the wood products (i.e., considering the loss of future income in the event of plantation failure) and/or from the provision of other services. Indeed, it may be that, in addition to the benefits mentioned above, replanting is more beneficial to the forest owner than not replanting, especially in the case of high mortality.

However, including these points in the analysis requires data that are either currently not available or that do not exist. More precisely on the second point, the computation of benefits requires prices and the assessment of timber production. Forest growth differs depending on the forest management implemented, which therefore influences final wood products. Estimating this timber production requires data that do not exist or forest growth models that do not consider different types of plantations. On the latter, forest growth models are poorly developed for these young stages or with many uncertainties due to difficulties in modeling recruitment dynamics [24].

## 5. Conclusions

Worldwide, the area of naturally regenerating forests has decreased since 1990 (at a declining rate of loss), but the area of planted forests has increased by 123 Mha, representing 7% of the forest area [25]. Planting is thus an important issue all over the world where large plantations are forecasted, especially in a context of climate change where carbon storage is an issue.

In France, since 2021, plantation is also largely encouraged through the "Plan de Relance" (https://agriculture.gouv.fr/francerelance-le-renouvellement-des-forets-francaises (accessed on 3 February 2023)) in which the government dedicated 200 M€ to reforestation programs. The stated objective is to plant 45,000 ha of forests in the coming years, leading to the sequestration of 150,000 additional tons of carbon each year. Such a plan will make it possible to increase planted surfaces, to regenerate existing forests, and to reconstitute those in decline.

While the State subsidizes plantation programs, the coverage of plantation failure remains an issue to be addressed and a risk to be minimized. From the forestry company's point of view, the problem lies in the increasing number of plantation failures where the company is liable even when it is not at fault. From the forest owner's point of view, the problem is also related to these increasingly likely plantation failures, with an ever more cumbersome risk. This research article is part of this debate.

We analyzed the costs of plantation failure and evaluated the distribution of replantation costs and risk sharing between the forestry company and the forest owner. Our results reveal that not replanting is less expensive for the forest owner than replanting or starting the whole plantation from scratch. In addition to this, reducing the forestry company's liability by increasing the deductible level corresponds to lower costs for the forestry company and higher ones for the forest owner. This additional risk level for the forest owner can be transferred to a private insurer.

This is one of the main conclusions of this article: the private insurance sector has an important role to play in the coverage of plantation failure. Further research needs to be conducted in this direction.

**Author Contributions:** Conceptualization, S.B.-A., M.B. and P.A.-D.; Methodology, S.B.-A.; Software, S.B.-A.; Validation, M.B. and P.A.-D.; Formal Analysis, S.B.-A.; Investigation, S.B.-A., M.B. and P.A.-D.; Resources, REPLANT-CLIC project; Data Curation, S.B.-A.; Writing—Original Draft Preparation, S.B.-A. and M.B.; Writing—Review & Editing, S.B.-A., M.B. and P.A.-D.; Visualization, S.B.-A.; Supervision, S.B.-A., M.B. and P.A.-D.; Project Administration, S.B.-A.; Funding Acquisition, S.B.-A., M.B. and P.A.-D. All authors have read and agreed to the published version of the manuscript.

**Funding:** This study benefits from a grant from the REPLANT-CLIC project, which was supported by the France Bois Forêt interprofessional organization and the French Ministry of Agriculture and Food Safety. The UMR BETA is supported by a grant overseen by the French National Research Agency (ANR) as part of the "Investissements d'Avenir" program (ANR-ll-LABX-0002–01, Lab of Excellence ARBRE).

**Data Availability Statement:** Not applicable.

**Acknowledgments:** This work was performed in partnership with Jonathan Pitaud and Clara Tallieu, whom we thank for our involvement in the project REPLANT-CLIC and their suggestions.

**Conflicts of Interest:** The authors declare no conflict of interest. The funders had no role in the design of the study; in the collection, analyses, or interpretation of data; in the writing of the manuscript; or in the decision to publish the results.

## Appendix A. Detailed Itinerary of Plantations for Douglas Fir and Sessile Oak

**Table A1.** Itinerary of plantations with the detailed operations associated with their year of realization and their costs in EUR for Douglas fir and sessile oak. The itinerary code corresponds to the species (DF for Douglas fir (*Pseudotsuga menziessi* Mirb.), SO for sessile oak (*Quercus petraea* Liebl.)), followed by the type of plants (CS for container seedlings, SBR for seedlings supplied with bare roots), the plantation type (C for plantation with a cane, P with a pickaxe, IP in individual pits), and game protection (F for fencing, IGP for individual protection, R for game repellent).

| Species | IT Code | Year | Operation | Stand Density | Price per Unit | Cost per ha |
|---|---|---|---|---|---|---|
| Douglas fir | DF_SBR-P+R | 1 | Preparation of the vegetation and the soil | | | €1250 |
| | | 1 | Plant with game repellent + plantation | 1600 | €1.25 | €2000 |
| | | 2 | Game repellent on survival | 1600 | €0.30 | €480 |
| | | 2 | Removal | | | €350 |
| | | 2 | Replantation: Plant + plantation | | €1.50 | |
| | | 3 | Removal | | | €350 |
| | | 7 | Maintenance | | | €350 |
| Douglas fir | DF_CS-C+R | 1 | Preparation of the vegetation and the soil | | | €1250 |
| | | 1 | Plant with game repellent + plantation | 1600 | €1.05 | €1680 |
| | | 2 | Game repellent on survival | 1600 | €0.30 | €480 |
| | | 2 | Removal | | | €350 |
| | | 2 | Replantation: Plant + plantation | | €1.50 | |
| | | 3 | Removal | | | €350 |
| | | 7 | Maintenance | | | €350 |
| Douglas fir | DF_SBR-IP+R | 1 | Preparation of the vegetation and the soil | 1300 | €1.20 | €1560 |
| | | 1 | Plant with game repellent + plantation | 1300 | €1.25 | €1625 |
| | | 2 | Game repellent on survival | 1300 | €0.30 | €390 |
| | | 2 | Removal | | | €350 |
| | | 2 | Replantation: Plant + plantation | | €1.50 | |
| | | 3 | Removal | | | €350 |
| | | 7 | Maintenance | | | €350 |

**Table A1.** *Cont.*

| Species | IT Code | Year | Operation | Stand Density | Price per Unit | Cost per ha |
|---|---|---|---|---|---|---|
| Douglas fir | DF_CS-IP+R | 1 | Preparation of the vegetation and the soil | 1300 | €1.20 | €1560 |
| | | 1 | Plant with game repellent + plantation | 1300 | €1.05 | €1365 |
| | | 2 | Game repellent on survival | 1300 | €0.30 | €390 |
| | | 2 | Removal | | | €350 |
| | | 2 | Replantation: Plant + plantation | | €1.50 | |
| | | 3 | Removal | | | €350 |
| | | 7 | Maintenance | | | €350 |
| Sessile oak | SO_SBR-P+F | 1 | Preparation of the vegetation and the soil | | | €1250 |
| | | 1 | Plant + plantation | 1600 | €2.10 | €3360 |
| | | 1 | Installation of fencing | | | €3500 |
| | | 2 | Removal | | | €400 |
| | | 2 | Replantation: Plant + plantation | | €2.35 | |
| | | 3 | Removal | | | €400 |
| | | 4 | Removal | | | €400 |
| | | 10 | Maintenance | | | €350 |
| Sessile oak | SO_CS-C+F | 1 | Preparation of the vegetation and the soil | | | €1250 |
| | | 1 | Plant + plantation | 1600 | €1.70 | €2720 |
| | | 1 | Installation of fencing | | | €3500 |
| | | 2 | Removal | | | €400 |
| | | 2 | Replantation: Plant + plantation | | €2.35 | |
| | | 3 | Removal | | | €400 |
| | | 4 | Removal | | | €400 |
| | | 10 | Maintenance | | | €350 |
| Sessile oak | SO_SBR-IP+F | 1 | Preparation of the vegetation and the soil | 1300 | €1.20 | €1560 |
| | | 1 | Plant + plantation | 1300 | €2.10 | €2730 |
| | | 1 | Installation of fencing | | | €3500 |
| | | 2 | Removal | | | €400 |
| | | 2 | Replantation: Plant + plantation | | €2.35 | |
| | | 3 | Removal | | | €400 |
| | | 4 | Removal | | | €400 |
| | | 10 | Maintenance | | | €350 |

**Table A1.** *Cont.*

| Species | IT Code | Year | Operation | Stand Density | Price per Unit | Cost per ha |
|---|---|---|---|---|---|---|
| Sessile oak | SO_CS-IP+F | 1 | Preparation of the vegetation and the soil | 1300 | €1.20 | €1560 |
| | | 1 | Plant + plantation | 1300 | €1.70 | €2210 |
| | | 1 | Installation of fencing | | | €3500 |
| | | 2 | Removal | | | €400 |
| | | 2 | Replantation: Plant + plantation | | €2.35 | |
| | | 3 | Removal | | | €400 |
| | | 4 | Removal | | | €400 |
| | | 10 | Maintenance | | | €350 |
| Sessile oak | SO_SBR-P+IGP | 1 | Preparation of the vegetation and the soil | | | €1250 |
| | | 1 | Plant + plantation | 1600 | €2.10 | €3360 |
| | | 1 | Installation of individual game protection | 1600 | €3.50 | €5600 |
| | | 2 | Removal | | | €400 |
| | | 2 | Replantation: Plant with individual game protection + plantation | | €3.35 | |
| | | 3 | Removal | | | €400 |
| | | 4 | Removal | | | €400 |
| | | 10 | Maintenance | | | €350 |
| Sessile oak | SO_CS-C+IGP | 1 | Preparation of the vegetation and the soil | | | €1250 |
| | | 1 | Plant + plantation | 1600 | €1.70 | €2720 |
| | | 1 | Installation of individual game protection | 1600 | €3.50 | €5600 |
| | | 2 | Removal | | | €400 |
| | | 2 | Replantation: Plant with individual game protection + plantation | | €3.35 | |
| | | 3 | Removal | | | €400 |
| | | 4 | Removal | | | €400 |
| | | 10 | Maintenance | | | €350 |

**Table A1.** *Cont.*

| Species | IT Code | Year | Operation | Stand Density | Price per Unit | Cost per ha |
|---|---|---|---|---|---|---|
| Sessile oak | SO_SBR-IP+IGP | 1 | Preparation of the vegetation and the soil | 1300 | €1.20 | €1560 |
| | | 1 | Plant + plantation | 1300 | €2.10 | €2730 |
| | | 1 | Installation of individual game protection | 1300 | €3.50 | €4550 |
| | | 2 | Removal | | | €400 |
| | | 2 | Replantation: Plant with individual game protection + plantation | | €3.35 | |
| | | 3 | Removal | | | €400 |
| | | 4 | Removal | | | €400 |
| | | 10 | Maintenance | | | €350 |
| Sessile oak | SO_CS-IP+IGP | 1 | Preparation of the vegetation and the soil | 1300 | €1.20 | €1560 |
| | | 1 | Plant + plantation | 1300 | €1.70 | €2210 |
| | | 1 | Installation of individual game protection | 1300 | €3.50 | €4550 |
| | | 2 | Removal | | | €400 |
| | | 2 | Replantation: Plant with individual game protection + plantation | | €3.35 | |
| | | 3 | Removal | | | €400 |
| | | 4 | Removal | | | €400 |
| | | 10 | Maintenance | | | €350 |

## Appendix B. Total Discounted Costs for Douglas Fir and Sessile Oak Considering no Replantation, Replantation and New Plantation

**Table A2.** Total discounted costs of the forest owner in EUR/ha for Douglas fir (*Pseudotsuga menziessi* Mirb.) (itinerary "DF_CS-IP+R") considering no replantation after drought shock ("Without RP"), replantation without insurance by the forestry company ("With RP and no INS"), replantation with insurance by the forestry company ("With RP and INS") with three deductibles ("D 20%", "D 50%", "D 80%"), and full replanting ("New plantation"), for different mortality rates.

| Mortality Rate | Without RP | With RP and no INS | With RP and INS (D 20%) | With RP and INS (D 50%) | With RP and INS (D 80%) | New Plantation |
|---|---|---|---|---|---|---|
| 0 | €4213 | €4213 | €4213 | €4213 | €4213 | |
| 0.1 | €4235 | €4423 | €4423 | €4423 | €4423 | €7103 |
| 0.2 | €4257 | €4632 | €4632 | €4632 | €4632 | €7125 |
| 0.3 | €4279 | €4842 | €4654 | €4842 | €4842 | €7147 |
| 0.4 | €4301 | €5051 | €4676 | €5051 | €5051 | €7169 |
| 0.5 | €4323 | €5261 | €4698 | €5261 | €5261 | €7191 |
| 0.6 | €4345 | €5470 | €4720 | €5283 | €5470 | €7213 |
| 0.7 | €4367 | €5679 | €4742 | €5305 | €5679 | €7235 |
| 0.8 | €4389 | €5889 | €4764 | €5326 | €5889 | €7257 |
| 0.9 | €4411 | €6098 | €4786 | €5348 | €5911 | €7279 |

**Table A3.** Total discounted costs of the forest owner in EUR/ha for sessile oak (*Quercus petraea* Liebl.) (itinerary "SO_CS-IP+F") considering no replantation after drought shock ("Without RP"), replantation without insurance by the forestry company ("With RP and no INS"), replantation with insurance by the forestry company ("With RP and INS") with three deductibles ("D 20%", "D 50%", "D 80%"), and full replanting ("New plantation"), for different mortality rates.

| Mortality Rate | Without RP | With RP and no INS | With RP and INS (D 20%) | With RP and INS (D 50%) | With RP and INS (D 80%) | New Plantation |
|---|---|---|---|---|---|---|
| 0 | €8546 | €8546 | €8546 | €8546 | €8546 | |
| 0.1 | €8555 | €8849 | €8849 | €8849 | €8849 | €15,682 |
| 0.2 | €8564 | €9152 | €9152 | €9152 | €9152 | €15,692 |
| 0.3 | €8574 | €9455 | €9161 | €9455 | €9455 | €15,701 |
| 0.4 | €8583 | €9758 | €9170 | €9758 | €9758 | €15,710 |
| 0.5 | €8592 | €10,061 | €9180 | €10,061 | €10,061 | €15,720 |
| 0.6 | €8602 | €10,364 | €9189 | €10,070 | €10,364 | €15,729 |
| 0.7 | €8611 | €10,667 | €9198 | €10,079 | €10,667 | €15,739 |
| 0.8 | €8620 | €10,970 | €9208 | €10,089 | €10,970 | €15,748 |
| 0.9 | €8630 | €11,273 | €9217 | €10,098 | €10,979 | €15,757 |

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
