# Peer review of "A Cost Assessment of Tree Plantation Failure under Extreme Drought Events in France: What Role for Insurance?"

_forests, doi:10.3390/f14020308_

Round 1
Reviewer 1 Report
The topic of this paper is impressive. The quality of the paper is quite good. The paper is interesting for journal readers. Kindly take note of the following specific comments to make it better.
1. The introduction needs improvement. The authors should reorganize the structure of the introduction section to thoroughly express the aspects of this study including the background, current progress, motivation, the research question, the objective, the contribution, etc.
2. I suggest that the present manuscript should highlight the novelty and the value of the work performed. The authors need to provide a strong argument here to show their novelty contributed to the field, the entire readership and the research community.
3. In the introduction section, the contribution should be expressed in a better way. In addition, expressions like firstly, secondly, and thirdly can be more useful in highlighting the contribution.
4. It is suggested to present the structure of the article at the end of the introduction.
5. Authors reviewed the previous literature; however, it will be good for the authors to create a paragraph at the end of the literature section to highlight the overall findings, gaps, and ways to fill this present study's gaps.
6. Authors should provide the steps of methodology in sequence.
7. The authors should present some research hypotheses as well as the discussion of the empirical results for the proposed research hypotheses.
8. It is suggested to avoid long paragraphs. It's confusing to the reader. The paragraph should be concise with clarity.
Reviewer 2 Report
Review of the paper “An economic analysis of tree plantation failure under extreme drought events: What role for insurance?” Submitted to Forests
The paper approaches a very relevant topic, as forests have an increasingly important role on the Planet balance and for a sustainable system. The authors focus on the case of plantation failure, caused by climatic extreme events like severe droughts, and its consequences in terms of costs sharing for forest owners and forest companies. Particularly, the authors try to find the best solution for forest companies, testing different cost sharing schemes (with different deductibles) and different options after the plantation failure: no replanting; replanting with distinct deductible levels and new plantation.
However, in my opinion, the paper presents several shortcomings that should first be solved, before being published in an International Journal like Forests (Q1). I will state below some of the main issues:
The authors underline the following risk sharing scheme in France:
“the forestry company provides a replacement for dead plants if the mortality exceeds 20% of the plantation. Mortality not covered by the insurance (20% in this case) remains the forest owner's liability.”
“However, the insurance can be waived if the mortality of the plants is unrelated to the quality of the forestry company's service, e.g., in periods of excessive drought or damage caused by animals, including intense browsing or infestations.”
As the authors are communicating with the international audience of “Forests” journal, the importance of this problem worldwide should be underlined right from the start:
- is this a common situation / Is it the same in other countries? All / some / few?
- This risk sharing procedure (this 20% …): what are the implications for both actors (forest companies and forest owners)? The paper should present numbers / data that could better explain the real relevance of the problem (cost and losses for both agents)
- forest companies and forest owners (public / private?) should be better described, as well as the stakeholders involved in forest management. The paper is based on a case study from France, so the French situation in terms of type of property / services provided by forest companies / regulation, etc. should be clear.
The first research question posed by the authors is: “In the case of a drought shock in the plantation, is it better, from an economic point of view, to replant, not to replant or to fully replant (i.e., starting the whole plantation from scratch)?”. This raises some questions that need to be clarified:
- authors should clarify / explicitly state, better for whom? For the forest company or for the forest owner? What about the society as a whole (other stakeholder’s interests), considering the ecosystem services provided by the forest?
- “from an economic point of view” – this should be defined. The paper should have a section devoted to economic analysis / assessment methods, to explain what is meant by this “economic point of view”, different ways of doing it and justification for the methodology adopted in the paper.
Additionally, the authors state:
“Morkovina et al compared two types of planting (traditional and innovative) through an economic assessment of their costs.”
and
“There is no study to date that assesses the economic aspects of plantation failure due to drought. In addition to this, the issue of shared liability between the forest owner and the forestry company has not yet been investigated.”
- this needs further development: what was the methodology used by Morkovina et al to do the economic assessment of costs?
- Again, what is meant by “economic aspects”? Are these only costs?? They shouldn’t be…
- what are the costs, in terms of losses for forest companies, due to the activation of this insurance clause? As this clause is part of the plantation contracts, a price premium should already be charged by forest companies to cover these situations. This gap in the literature pointed by the authors (inexistence of studies about shared liability) should be further justified: is it because the problem isn’t real / serious? Is it because it consists of a problem that only exists in France? ….
The relevance of some results presented is questionable:
- The authors identify the itinerary that minimized the plantation costs for each species (based on a TDC, with a discount rate of 2%., which is not justified). This is independent from the research question: these practices to minimize plantation costs should be used independently of any climate extreme event or insurance scheme….
- The authors state:
“In this article, we analyze the economic impacts of plantation failure and evaluate the distribution of economic costs between the forestry company and the forest owner. To do this, we perform an economic evaluation of changes in planting practices for the three main species of plantations in France.” What did the authors mean by changes in planting practices? The authors considered several options for the optimal plantation scheme (CS, SBR, etc….) but their recommendation was independent on this: replanting / no replanting, etc.
- Table 5 is at the centre of the paper recommendations and it should be better justified / explained: how did the authors computed all the costs for different mortality rates? Why higher costs of new plantation with zero mortality rate (does it make sense a new plantation in that case)?
Furthermore, an economic assessment for the options of plantation / no replantation of the dead plants cannot merely include the costs of plantation: what is the compensation for the forest owner in the case of no replantation? The forest company do not support the costs of replanting but the owner loses the revenues from the wood…
The authors state:
“Here, the deductible of 20% implies that the forestry company bears most of the risk. Increasing the deductible means a lower liability for the forestry company and more liability for the forest owner. This is why we tested two other deductible levels that imply a higher risk borne by the forest owner: 50%, and 80%.”
However, insurance companies offer different insurance schemes, with different deductibles but also with different price premiums, i.e., the least the deductible, the higher the price premium. This would obviously be the same for the forest company: having higher deductibles should also imply a lower price for the plantation service (in comparison to the current price, with the 20% deductible). And this aspect seems to be completely ignored by the authors.
At the Discussion section, authors highlight that the main results show that no replanting is less expensive for the forest owner than replanting or full replanting.
For mortality rate equals to 0, the forest owner supports only the initial cost of plantation 1659€. It should be explained why is the “no replanting” costs increasing with the mortality rate. The paper presents the way total discounted costs were calculated (TDC) for the different itineraries only for the case without mortality rate. Authors must better explain how mortality rate was included at TDC calculation: the formula must be presented.
Authors state: “increasing the deductible results in a higher cost reduction for the forestry company than the cost increase for the forest owner.” Again TDC computation with mortality must be better explained: increasing the deductible obviously increases replantation costs for forest owners and decreases for forest companies. However, those different insurance schemes should also have impact on the plantation costs (the lower the deductible the higher the plantation cost – price premium - and vice versa).
“In this context, we identified two ways of solving the forestry company’s problem of financial instability related to increasing costs due to plantation failure.”
Was this the paper’s goal? If so, it should be clear from the start that the main concern is about the forest company (and also the paper title should reflect this). Also, the “problem of financial instability” should have been presented at the beginning of the paper, introducing numbers / data that could support this problem / reality. Authors should introduce in the literature revision studies presenting results that prove the correspondence between the forementioned financial instability of forest companies and losses related with severe drought periods.
Also, the authors should justify why focusing on forest companies instead of forest owners (as one of the solutions being presented – increasing the deductible – favours forest companies but increases forest owners costs). The paper should introduce a section where forest main stakeholders are presented.
“having a reliable drought index may solve the problem. (…) . As soon as the plantation failure is clearly identified as a consequence of a drought event (through a reliable index), the forestry company's liability will not be engaged. In that case, the costs associated with the plantation failure would be borne by the forest owner. (…) including specific clauses in the replanting contracts that explicitly exclude the forestry company’s liability would result in a reduction in the forestry company’s costs.”
I totally agree with this suggestion to solve the forestry companies financial problems, as this is the most obvious answer for the problem stated within the paper. However, this raises the question about the added value of the present study, as this conclusion could be reached without any of the results presented.
“Taking the services provided by forests to tackle climate change challenges (carbon storage, substitution of fossil fuel-based products by wood products) into account also reveals the need to replant.”
This is a major point that was not included in the authors calculations: the value of the ecosystem services provided by forests. This is a considerable shortcoming of the method used by the authors to account for an economic analysis of tree plantation failure.
Although the authors recognize this could be and improvement / future research, from my point of view, only a cost-benefit analysis would make sense here (so as to include total costs and benefits, as well as benefit losses that occur with plantation failure and with the decision of no replanting). The authors say that “including these points in the analysis requires data that are either not available or that do not exist”. That is not exact as there are ways to estimate the loss revenues per ha (each of the 3 species considered have a market value, a mean selling price).
The real problem here is the insurance scheme, and the fact that the forest company can be liable for mortality of plants even if that mortality is caused by non-controled events, extreme events like droughts or others. So, the real contribution here should be on the insurance clauses and definition of extreme droughts, as the authors recognize in the paper:
“In view of the reforestation sector's difficulties in reducing the costs related to excess mortality, changing the clause governing insurance services for the success/survival of plantations in the event of extreme drought episodes can allow the forestry company to be released from its liability.”
In sum, major improvements in the paper and correction of some major shortcomings are necessary to clarify its relevance before being published.
Round 2
Reviewer 1 Report
Comments are well addressed. Recommended for publication.
Reviewer 2 Report
Thank you for your careful answers to my comments and suggestions. You did a great job with the revision of the paper, that I consider to be much clearer now and interesting for Forest readers. Congratulations for your paper and research!